# Yeast Fin1-PP1 dephosphorylates an Ipl1 substrate, Ndc80, to remove Bub1-Bub3 checkpoint proteins from the kinetochore during anaphase

Michael Bokros[1,2☯], Delaney Sherwin[1☯], Marie-Helene Kabbaj[1], Yanchang Wang[1]*

**1** Department of Biomedical Sciences, College of Medicine, Florida State University, Tallahassee, Florida, United States of America, **2** Sylvester Comprehensive Cancer Center, University of Miami, Miami, Florida, United States of America

☯ These authors contributed equally to this work.
* yanchang.wang@med.fsu.edu

**Data Availability Statement:** All relevant data are within the manuscript and its Supporting Information files.

## Abstract

The spindle assembly checkpoint (SAC) prevents anaphase onset in response to chromosome attachment defects, and SAC silencing is essential for anaphase onset. Following anaphase onset, activated Cdc14 phosphatase dephosphorylates the substrates of cyclin-dependent kinase to facilitate anaphase progression and mitotic exit. In budding yeast, Cdc14 dephosphorylates Fin1, a regulatory subunit of protein phosphatase 1 (PP1), to enable kinetochore localization of Fin1-PP1. We previously showed that kinetochore-localized Fin1-PP1 promotes the removal of the SAC protein Bub1 from the kinetochore during anaphase. We report here that Fin1-PP1 also promotes kinetochore removal of Bub3, the Bub1 partner, but has no effect on another SAC protein Mad1. Moreover, the kinetochore localization of Bub1-Bub3 during anaphase requires Aurora B/Ipl1 kinase activity. We further showed that Fin1-PP1 facilitates the dephosphorylation of kinetochore protein Ndc80, a known Ipl1 substrate. This dephosphorylation reduces kinetochore association of Bub1-Bub3 during anaphase. In addition, we found that untimely Ndc80 dephosphorylation causes viability loss in response to tensionless chromosome attachments. These results suggest that timely localization of Fin1-PP1 to the kinetochore controls the functional window of SAC and is therefore critical for faithful chromosome segregation.

## Author summary

Mistakes in chromosome attachment activate the spindle assembly checkpoint to stop the cell cycle for error correction. This activation depends on kinetochore recruitment of checkpoint proteins, including the Bub1-Bub3 complex. After cells have established chromosome bipolar attachment, the spindle assembly checkpoint is silenced by protein phosphatase 1 (PP1) to enable anaphase entry. Fin1 is a regulatory subunit of PP1 in budding yeast, and Fin1 recruits PP1 to the kinetochore during anaphase when the spindle

**Funding:** Y.W. received support (grarnt no R01GM121786) from the National Institute of Health. The funders had no role in study design, data collection and analysis, decision to publish, or preparation of the manuscript.

**Competing interests:** The authors have declared that no competing interests exist.

assembly checkpoint is silenced. We previously showed that Fin1-PP1 promotes the dissociation of checkpoint protein Bub1 from the kinetochore during anaphase, revealing an unexpected regulation of the spindle assembly checkpoint after its silencing. Here, we showed that Fin1-PP1 promotes the kinetochore dissociation of both Bub1 and Bub3 during anaphase, and that the kinetochore association of Bub1 and Bub3 is interdependent. It has been shown that PP1 antagonizes the phosphorylation imposed by Ipl1/Aurora B, a conserved protein kinase critical for accurate chromosome segregation. We further found that Fin1-PP1 promotes the dephosphorylation of kinetochore protein Ndc80, an Ipl1 substrate, which reduces kinetochore association of Bub1-Bub3. Moreover, dysregulation of Ndc80 phosphorylation results in increased viability loss when syntelic attachment is induced. Therefore, we identified a new layer of regulation of the spindle assembly checkpoint during anaphase, which ensures its functional window during cell cycle.

## Introduction

During cell division, chromosomes are segregated equally into daughter cells, and faithful chromosome segregation is essential to maintain the genome integrity of all living organisms. The failure of this process results in aneuploidy, a hallmark of cancers and genetic diseases like Trisomy 21 [1,2]. During metaphase, chromosomes are attached to microtubules through their kinetochores in a bi-oriented fashion to generate tension. Incorrect attachments between the kinetochore and the spindle microtubules activate the spindle assembly checkpoint (SAC) to prevent anaphase onset. The SAC activation depends on the recruitment of SAC proteins to kinetochores [3]. In addition, SAC activation relies on a delicate balance between kinase and phosphatase activity. The phosphorylation of the MELT motifs in kinetochore protein Spc105 by SAC kinase Mps1 promotes its association with SAC proteins Bub3 and Bub1. Bub1 further recruits SAC proteins Mad1 and Mad2 for SAC activation [4–6].

In budding yeast, tensionless attachments delay anaphase onset by preventing SAC silencing, and this delay depends on Ipl1, the homologue of human Aurora B kinase [7,8]. Ipl1[Aurora-B], together with Sli15[INCENP], Bir1[Survivin] and Nbl1[Borealin], constitutes the Chromosomal Passenger Complex (CPC). Our previous work indicates that Ipl1 kinase prevents SAC silencing in the presence of tensionless attachments through the phosphorylation of a kinetochore protein Dam1 [9,10]. In addition, Dam1 phosphorylation destabilizes kinetochore-microtubule interaction. Therefore, Ipl1-dependent phosphorylation of Dam1 not only facilitates correction of attachment errors but also prevents premature SAC silencing. Although SAC silencing requires an increase in phosphatase activity at the kinetochore, the mechanisms that control the kinase/phosphatase balance during cell cycle remain unclear. It is speculated that the tension generated by chromosome bipolar attachment triggers a balance change for SAC silencing [11,12].

SAC silencing allows anaphase onset, which triggers further cellular change of kinase/phosphatase balance. In budding yeast, the activity of phosphatase Cdc14 has been shown to be increased in anaphase to antagonize cyclin dependent kinase (CDK). The activity of Cdc14 is regulated through its subcellular localization. Prior to anaphase onset, Cdc14 is sequestered within the nucleolus by binding to nucleolar proteins Net1/Cfi1 and Tof2 [13–15]. Following anaphase onset, Cdc14 is transiently released from the nucleolus, and this process is controlled by the Cdc14 early anaphase release (FEAR) pathway. FEAR-dependent Cdc14 release reverses the phosphorylation imposed by S-phase CDK to promote early anaphase events [16,17].

Because about a dozen S-phase CDK substrates have been identified [18], more work is needed to understand the function for the dephosphorylation of each substrate.

Recently we found that Fin1, a regulatory subunit of protein phosphatase 1 (PP1), is modulated by the CDK-Cdc14 axis to further suppress SAC during anaphase [19,20]. S-phase CDK phosphorylates Fin1 to promote its binding to Bmh1 and Bmh2, which sequester Fin1 and prevent its kinetochore localization [21,22]. Upon anaphase onset and FEAR activation, Cdc14 dephosphorylates Fin1 and allows its robust kinetochore binding. We previously found that in the absence of Fin1, the SAC protein Bub1 remains at the kinetochore during anaphase, indicating that Fin1-PP1 functions to remove SAC proteins during anaphase [19]. Conversely, premature kinetochore localization of Fin1-PP1 likely compromises SAC activity and results in sensitivity to the induction of chromosome syntelic attachments, a condition where both sister kinetochores are attached by microtubules from the same spindle pole. Further supporting this idea, activation of Cdc14 phosphatase, which reverses the phosphorylation imposed by CDK, also prevents SAC reactivation in budding yeast [23]. However, the molecular mechanism by which Fin1-PP1 regulates SAC activity remains elusive.

Here we have further investigated how Fin1-PP1 complex promotes the removal of SAC proteins from the kinetochore during anaphase. First, we found that, in addition to Bub1, the Fin1-PP1 complex also promotes the kinetochore removal of Bub1's partner Bub3 during anaphase. Moreover, the kinetochore localization of Bub1 and Bub3 is interdependent, suggesting that the formation of Bub1-Bub3 complex is critical for their kinetochore localization. In addition to Dam1, Ipl1 kinase also phosphorylates Ndc80, a subunit of a kinetochore complex that interacts with both the inner kinetochore and microtubule-associated Dam1 complex [24]. We showed that Fin1-PP1 promotes the dephosphorylation of Ndc80 to prevent Bub1-Bub3 re-association with the kinetochore during anaphase. Like cells with enhanced Fin1 kinetochore localization, a phospho-deficient mutation in Ndc80 also causes viability loss in cells with syntelic chromosome attachment. Together, our research work has revealed the mechanism of SAC protein removal from the kinetochore by Fin1-PP1 after SAC silencing. Exposing this mechanism demonstrates the significance of the tight control of kinase-phosphatase balance at the kinetochore in faithful chromosome segregation.

## Results

### Fin1-PP1 prevents the reassociation of SAC proteins Bub1 and Bub3 with the kinetochore in anaphase

We previously found that *fin1Δ* mutant cells retained kinetochore localization of SAC protein Bub1 during anaphase [19]. However, it remains unclear if the Bub1 kinetochore localization in *fin1Δ* mutant cells during anaphase is persistent or dynamic. Therefore, we followed Bub1 kinetochore localization using live-cell imaging in wild-type (WT) and *fin1Δ* cells expressing Bub1-GFP and an established kinetochore marker Nuf2-mCherry [25]. At the time of anaphase onset, both WT and *fin1Δ* cells hardly showed Bub1 kinetochore localization. Following anaphase onset, only 7% of WT cells displayed weak Bub1 kinetochore localization at one or more time frames during anaphase. Strikingly, 91% of *fin1Δ* mutant cells displayed Bub1-GFP at the kinetochore during anaphase (Fig 1A). Additionally, the localization of Bub1 to the kinetochore during anaphase was dynamic in *fin1Δ* mutant cells, alternating between association and dissociation throughout anaphase. The kinetochore association of Bub1 occurred in 35% of all frames following anaphase onset in *fin1Δ* cells. Moreover, we noticed that Bub1 kinetochore localization was not evenly distributed between the two kinetochore clusters in *fin1Δ* cells, supporting the idea that Bub1 kinetochore localization in *fin1Δ* mutant cells during

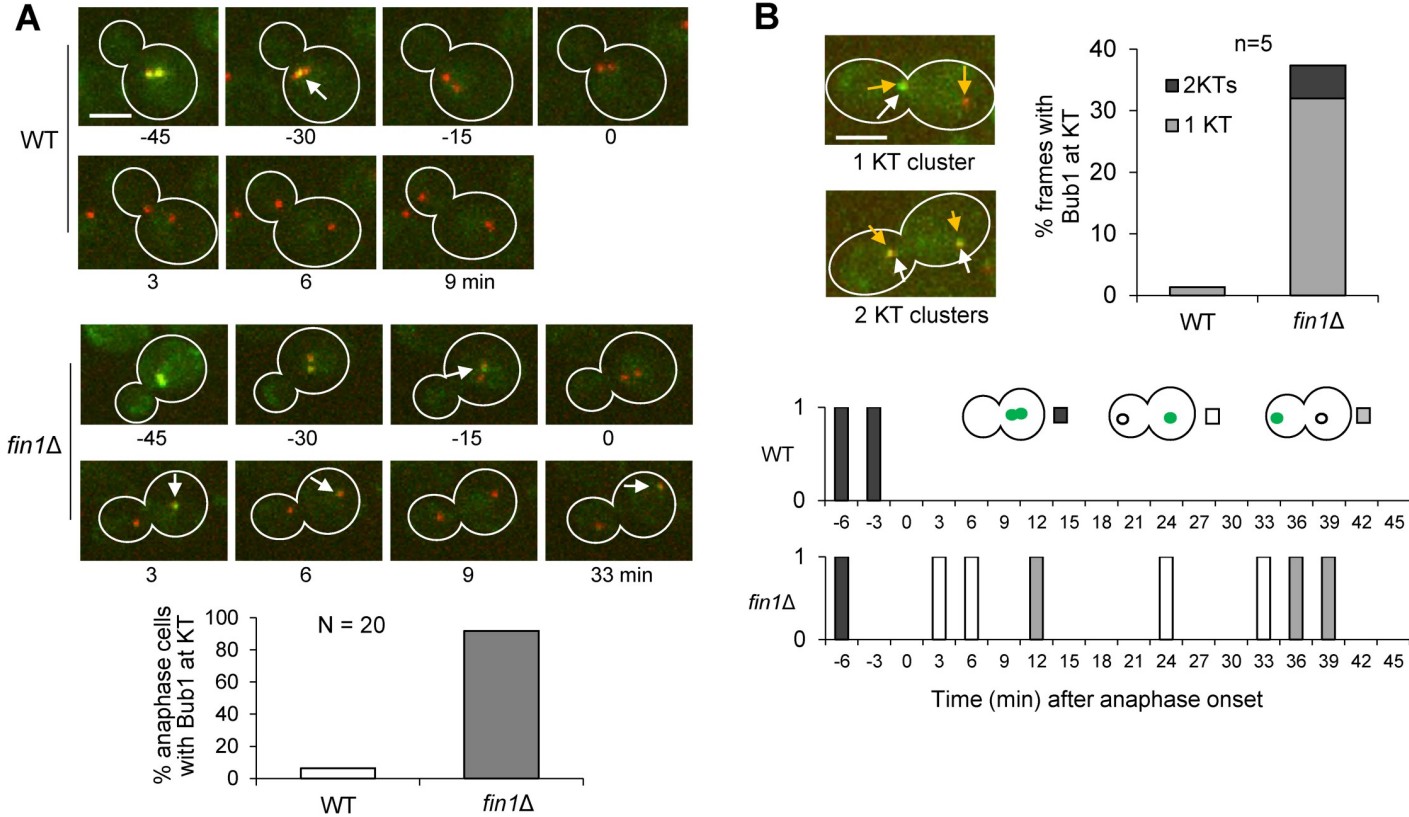

**Fig 1. Fin1 promotes Bub1 dissociation from the kinetochore during anaphase.** (A) Live-cell imaging of WT (2827-1-4) and *fin1Δ* (3196-1-3) cells to show association of Bub1 with the kinetochore in anaphase. Log-phase WT and *fin1Δ* cells with Bub1-GFP and Nuf2-mCherry in yeast peptone dextrose (YPD) medium were spotted onto the surface of a slide covered with agarose medium and subjected to live-cell microscopy. We set time 0 as the last point when the distance between two Nuf2-mCherry foci is less than 3μm. Arrows indicate Bub1-GFP Nuf2-mCherry co-localization. Cells that had Bub1-GFP co-localization with Nuf2-mCherry at any time during anaphase was counted as positive and the percentage of positive cells was graphed (n = 20). KT: kinetochore. (B) The co-localization of Bub1-GFP with the kinetochore after anaphase onset is dynamic. The localization pattern of Bub1 with single or both kinetochore clusters was counted for representative WT and *fin1Δ* cells during anaphase (n = 5). Yellow arrows indicate kinetochore cluster; white arrows indicate Bub1 kinetochore localization. Scale bar, 2.5μm. The presence (1) or absence (0) of co-localization between Bub1-GFP and Nuf2-mCherry through the time course is shown in the bottom right panel.

anaphase is a dynamic process (Fig 1B). We reason that Fin1-PP1 prevents kinetochore re-association of SAC protein Bub1 after anaphase entry.

## Fin1-PP1 prevents anaphase kinetochore localization of both Bub1 and Bub3

The first step of SAC assembly at the kinetochore is the association of Bub3 with Spc105 MELT motifs after the phosphorylation of Spc105 by Mps1 kinase [4,6]. Bub1 follows, binding to Bub3 through its Gle2-binding sequence [26,27]. Because we showed the retention of Bub1 on ana-phase kinetochores in *fin1Δ* cells [19], we also tracked the kinetochore localization of Bub3 in *fin1Δ* mutant cells. We used a temperature-sensitive *cdc15-2* mutant to arrest the cells in telo-phase, and examined the co-localization of Bub3-GFP with the kinetochore marker Nuf2-m-Cherry. In cells arrested by *cdc15-2*, the frequency of Bub3 kinetochore localization was as low as 1%. However, under these same conditions, 96% of *fin1Δ* cells showed Bub3 kinetochore locali-zation (Fig 2A). Thus, *fin1Δ* cells show retention of both Bub1 and Bub3 at the kinetochore when arrested at telophase by *cdc15-2*. We also performed live-cell imaging using strains with Bub3-GFP and Nuf2-mCherry and observed increased Bub3-kinetochore association in *fin1Δ*

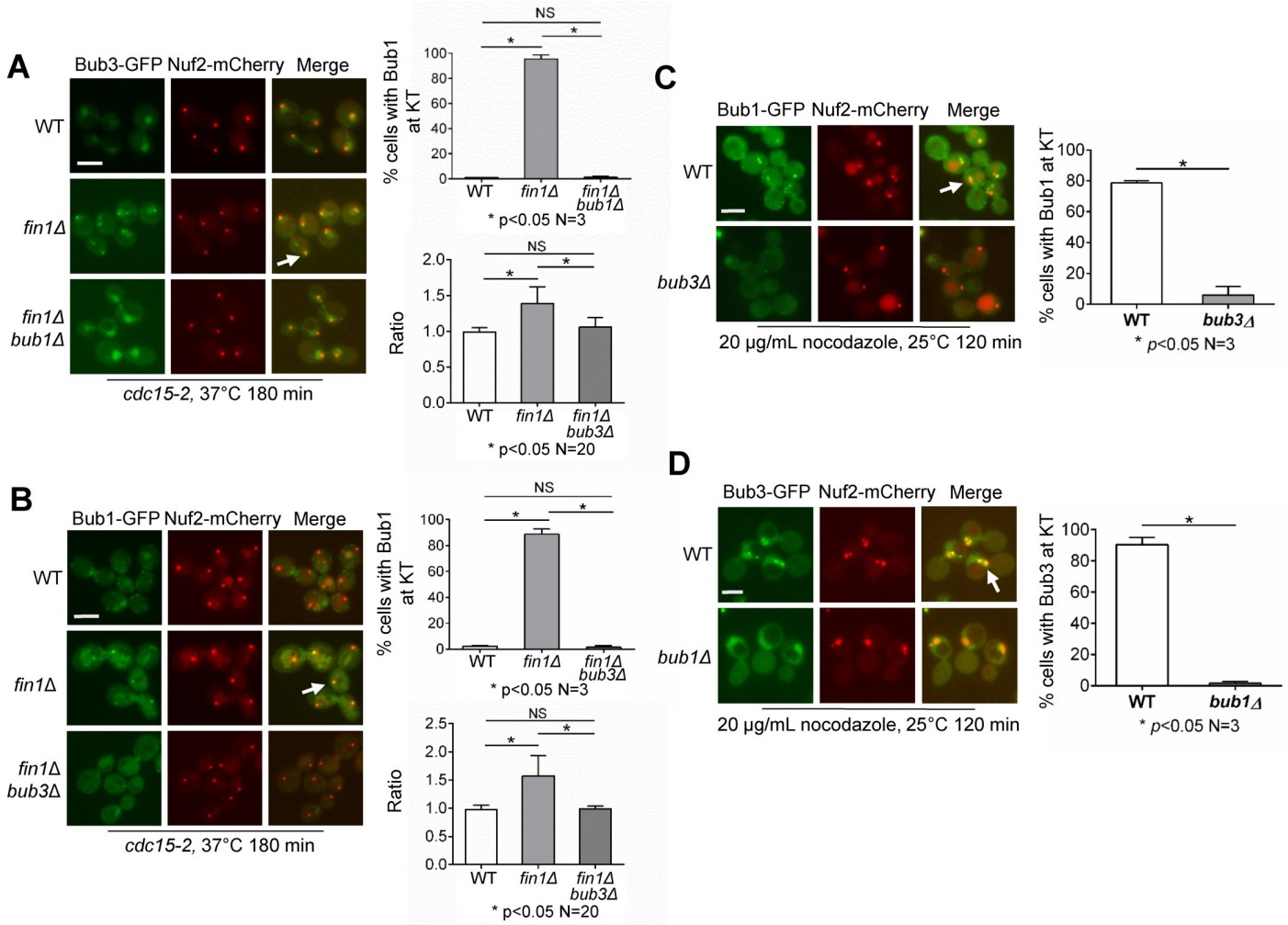

**Fig 2. The kinetochore localization of Bub1 and Bub3 in anaphase is interdependent. (A)** Bub3 shows kinetochore localization in *fin1Δ* cells during anaphase and this localization depends on Bub1. WT (3888-11-2), *fin1Δ* (3890-14-3), and *fin1Δ bub1Δ* (4065-4-3) cells in a *cdc15-2* background containing Bub3-GFP and Nuf2-mCherry were grown to log-phase at 25˚C. The cells were then shifted to the non-permissive temperature of 37˚C for 180 minutes to allow inactivation of Cdc15. Samples were collected and pictures taken. The frequency of co-localization of Bub3-GFP and Nuf2-mCherry signals was counted for each strain (n = 100). White arrows indicate Bub3-kinetochore co-localization. The experiment was repeated three times and statistical significance determined by $p < 0.05$ using Kruskal-Wallis one-way ANOVA. Scale bar, 5 μm. The ratio of Bub3-GFP intensity at the kinetochore over other nuclear regions was also measured using ImageJ (n = 20). The average ratio is shown in the bottom right panel. Statistical significance determined by $p < 0.05$ using Kruskal-Wallis one-way ANOVA. **(B)** The kinetochore localization of Bub1 in anaphase *fin1Δ* cells depends on Bub3. WT (3177-3-4), *fin1Δ* (3196-3-1), and *fin1Δ bub3Δ* (3905-6-1) cells in a *cdc15-2* background containing Bub1-GFP and Nuf2-mCherry were grown as described above. The frequency of co-localization of Bub1-GFP and Nuf2-mCherry was counted for each strain (n = 100). White arrows indicate Bub1-kinetochore co-localization. The experiment was repeated three times and statistical significance was determined as described above. Scale bar, 5 μm. The ratio of Bub1-GFP intensity at the kinetochore over other nuclear regions was also measured using ImageJ (n = 20). The average ratio is shown in the bottom right panel. Statistical significance determined by $p < 0.05$ using Kruskal-Wallis one-way ANOVA. **(C)** The kinetochore localization of Bub1 in nocodazole-treated cells depends on Bub3. WT (3887-4-1) and *bub3Δ* (3887-3-4) cells with Bub1-GFP and Nuf2-mCherry were grown to log-phase at 25˚C. Nocodazole was added to the cell cultures to a final concentration of 20 μg/mL. After incubation for 120 minutes, samples were collected and pictures taken. The experiment was repeated three times and statistical significance was determined. Scale bar, 5 μm. **(D)** The kinetochore localization of Bub3 in nocodazole-treated cells depends on Bub1. WT (4065-8-1) and *bub1Δ* (4065-3-1) cells with Bub3-GFP and Nuf2-mCherry were treated as in (C). The experiment was repeated three times and statistical significance determined by $p < 0.05$ using Wilcoxon rank sum test. Scale bar, 5 μm.

cells (S1A Fig). Only 5% of WT cells showed kinetochore Bub3 localization in anaphase, compared to 85% of *fin1Δ* mutant cells (S1B Fig). Bub3 also showed dynamic kinetochore localization during anaphase as 39% frames of *fin1Δ* cells during anaphase showed Bub3 kinetochore localization, compared to 2% for WT cells (S1C Fig). We recognize that the number of *fin1Δ* cells showing Bub1 or Bub3 kinetochore localization when arrested with *cdc15-2* is notably

higher than the frequency presented in the live-cell imaging data. We reason that in *cdc15-2* mutant cells, inactivation of the mitotic exit network may exacerbate the *fin1Δ* phenotype.

Although strong kinetochore association of Bub1-Bub3 complex was noticed in *cdc15-2 fin1Δ* mutant cells in anaphase, no Mad1 was detected at the anaphase kinetochore in *cdc15-2 fin1Δ* or asynchronous *fin1Δ* anaphase cells (S2 Fig). In these cells, Mad1 showed a characteristic nuclear envelop localization as described [28]. Thus, a distinct mechanism controls Mad1 kinetochore dissociation, which is likely the key to SAC silencing.

## The kinetochore association of Bub1 and Bub3 is interdependent

Earlier findings suggest that Bub3 is the initial binding protein to the kinetochore during SAC activation, and that Bub1 binds to Bub3 subsequently [27,29,30]. To test the dependency of Bub1 anaphase kinetochore localization on Bub3, we examined Bub1 kinetochore localization in *fin1Δ bub3Δ* mutant cells arrested with *cdc15-2*. In telophase-arrested cells, the frequency of Bub1 kinetochore localization was around 1%, whereas in *fin1Δ* cells, it was around 90% as expected. However, Bub1 anaphase kinetochore localization was completely abolished in *fin1Δ bub3Δ* double mutants (Fig 2B). We also measured the ratio of Bub1-GFP intensity at the kinetochore relative to other regions in the nucleus. This ratio for *cdc15-2* cells was close to one, indicating no kinetochore enrichment of Bub1, however the ratio increased to 1.4 in *cdc15-2 fin1Δ* cells indicating kinetochore enrichment of Bub1. This enrichment was abolished in *cdc15-2 bub3Δ fin1Δ* cells, suggesting this enrichment depends on Bub3 (Fig 2B). Using a similar protocol, we also examined the dependency of anaphase kinetochore localization of Bub3 on Bub1. In *fin1Δ bub1Δ* mutant cells, the anaphase kinetochore localization of Bub3 was completely abolished as well, with only 1% cells showing Bub3 kinetochore localization (Fig 2A). Moreover, Bub1-dependent enrichment of Bub3 at the kinetochore was supported by the ratio of Bub3-GFP intensity at the kinetochore relative to other regions in the nucleus (Fig 2A). These results indicate the interdependency of Bub1 and Bub3 for their anaphase kinetochore localization in *fin1Δ* mutants.

Given the interdependence of Bub1 and Bub3 for their kinetochore localization during anaphase in *fin1Δ* mutants, we sought to know if this interdependency exists in cells treated with nocodazole, which depolymerizes microtubules and disrupts kinetochore-microtubule interaction. Yeast cells treated with nocodazole show SAC activation and accumulation of SAC proteins on the kinetochore [31]. After treatment with 20 μg/mL nocodazole, 79% of WT yeast cells showed kinetochore localization of Bub1, but this localization decreased to only 6% in *bub3Δ* mutant cells (Fig 2C). In addition, we examined Bub3 kinetochore localization in *bub1Δ* cells under the same condition. In WT cells treated with nocodazole, 90% showed Bub3 kinetochore localization. Interestingly, eliminating Bub1 almost completely abolished the kinetochore localization of Bub3, down to 2% (Fig 2D). Because SAC activation is abolished in *bub1Δ* and *bub3Δ* mutants treated with nocodazole, we performed the experiment in cells without nocodazole treatment and examined the kinetochore localization of Bub1/Bub3 in metaphase cells. Similarly, Bub1 and Bub3 showed interdependent kinetochore localization (S3 Fig). One possible explanation is that the complex formation of Bub1-Bub3 is required for their stable kinetochore localization. We noticed that this result is inconsistent with published work showing that kinetochore localization Bub3 is independent of Bub1 and other checkpoint proteins [26]. Further experiments are required to clarify this discrepancy.

## The timing of Fin1-PP1 kinetochore localization is critical for proper chromosome segregation after exposure to spindle poisons

Fin1-PP1 is recruited to the kinetochore after anaphase entry [22]. However, it is unclear whether the kinetochore recruitment of Fin1-PP1 is sufficient to remove SAC proteins from

the kinetochore to inactivate the SAC. Fin1 is a substrate of S-phase CDK and Cdc14 phosphatase, and mutation of the CDK consensus sites (S36A, S54A, T68A, S117A, S148A) in *FIN1* gene generates a *fin1-5A* mutant. Mutated Fin1-5A localizes to the kinetochore prematurely [19,21]. Because Cdc14 phosphatase dephosphorylates Fin1 to allow kinetochore localization, we first examined the kinetochore localization of Fin1 and Fin1-5A in temperature-sensitive *cdc14-2* mutant cells. Fin1-5A, but not Fin1 showed kinetochore localization in *cdc14-2* cells grown at 37˚C, indicating that the kinetochore localization of Fin1-5A bypasses the requirement of Cdc14 (S4 Fig). We previously showed that *fin1-5A* cells were sensitive to the induction of syntelic attachment and to the spindle poisons benomyl and nocodazole [19,22]. Thus, we also examined the kinetochore localization of Fin1-5A in cells treated with nocodazole. We found the 83% yeast cells showed kinetochore localization of Fin1-5A-GFP when treated with nocodazole, while the frequency was only 10% for Fin1-GFP. Notably, no significant cell rebudding was observed for *fin1-5A* cells after $G_1$ release into nocodazole medium for 120 minutes (Fig 3A), which is different from the response of SAC mutants to nocodazole.

To test if kinetochore recruitment of Fin1-PP1 is sufficient to displace SAC proteins from the kinetochore, we analyzed Bub1 kinetochore localization in *fin1Δ BUB1-GFP NUF2-m-Cherry* cells carrying either WT *FIN1* or *fin1-5A* plasmid. Nocodazole-treated cells with either *FIN1* or *fin1-5A* plasmid showed high frequency of Bub1-kinetochore localization at 91% and 93%, respectively (Fig 3B). This result suggests that premature kinetochore association of Fin1-PP1 is not sufficient to remove Bub1-Bub3 from the kinetochore. To further assess SAC activity in *fin1-5A* cells, the degradation kinetics of Pds1 was analyzed. Pds1 is an anaphase inhibitor and SAC activation prevents Pds1 degradation [32]. Pds1 levels were measured in synchronized *FIN1* and *fin1-5A* cells treated with 20 μg/mL nocodazole. In WT cells, Pds1 levels increased and remained consistent. Similarly, *fin1-5A* cells also showed persistent Pds1 levels in the presence of nocodazole, suggesting SAC activation (Fig 3C). We further questioned if over a longer time period, the SAC in *fin1-5A* may eventually be weakened. To test this, we followed spindle elongation dynamics for 5 hours in temperature sensitive *ctf13-30* mutant cells which arrest at metaphase due to impaired kinetochore function [33]. Interestingly, *ctf13-30* cells carrying either *FIN1* or *fin1-5A* plasmid presented with short spindles, indicating metaphase arrest [34] (S5A Fig). This implies that premature Fin1-PP1 kinetochore localization is not sufficient to inactivate the SAC when kinetochore attachment was disrupted. One explanation is that SAC activation outweighs the negative regulation of SAC by Fin1-PP1 in cells with kinetochore attachment defects.

Our result indicates that *fin1-5A* cells exhibit intact SAC activity in response to nocodazole treatment or defective kinetochore function. To further understand the sensitivity of *fin1-5A* mutants to spindle poisons, we followed chromosome segregation in cells with GFP-marked centromere of chromosome IV (*CEN4*-GFP) and Tub1-mCherry after nocodazole exposure. $G_1$-synchronized cells with plasmids containing *FIN1* or *fin1-5A* were released into nocodazole (20μg/ml) for 2 hours. These cells were arrested efficiently by nocodazole as large budded cells. After these cells were spread onto YPD plates, 85% of *FIN1* cells were viable while only 40% of *fin1-5A* cells were viable (Fig 3D). We also followed *CEN4*-GFP segregation in these cells. After nocodazole washout for 10 minutes, spindle formation was observed. After 40 minutes, some cells showed an elongated spindle, and 16% of *fin1-5A* cells with an anaphase spindle exhibited *CEN4*-GFP missegregation, while this number was only 3% in *FIN1* cells (Fig 3D). We repeated the experiment using cells without nocodazole treatment, and no chromosome missegregation or viability loss was observed for cells expressing *fin1-5A* (S5B Fig). These results indicate that the premature kinetochore localization of Fin1-5A does not impair SAC activation in response to nocodazole treatment, but leads to higher frequency of chromosome missegregation during the recovery from nocodazole arrest. Untimely kinetochore

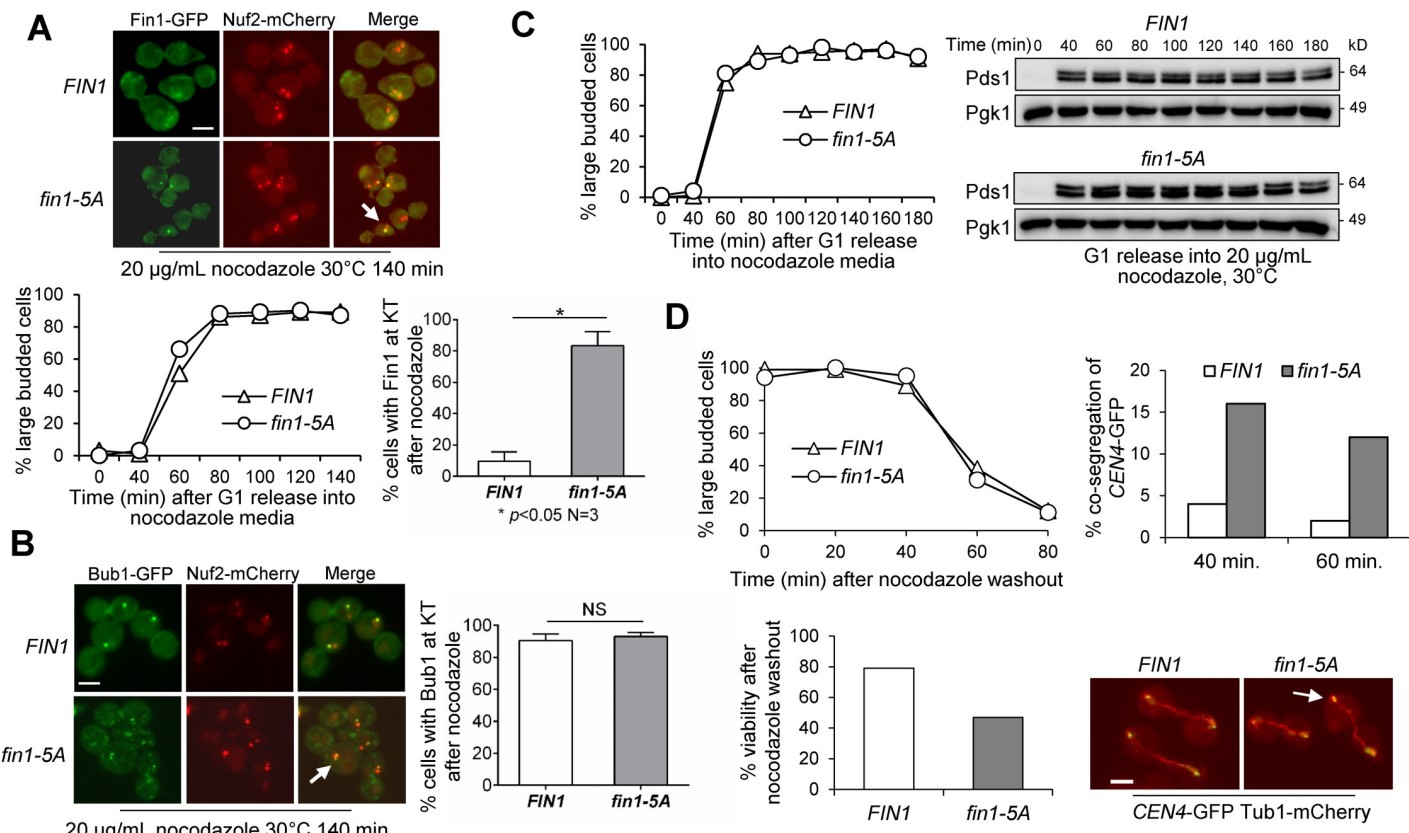

**Fig 3. The SAC is intact in response to nocodazole treatment in cells with premature Fin1-kinetochore localization. (A)** Premature kinetochore association of phospho-deficient Fin1-5A proteins in nocodazole-arrested cells. $G_1$-arrested cells carrying *FIN1-GFP* (pSB1252) or *fin1-5A-GFP* (pSB1359) plasmids and Nuf2-mCherry (3175-1-4) were released into YPD containing 20 μg/mL nocodazole for 140 min at 30˚C. Cells were collected and fixed. Budding index and Fin1-GFP kinetochore localization were counted (n = 100). White arrows indicate kinetochore localization of Fin1. The experiment was repeated three times and statistical significance determined by $p < 0.05$ using Wilcoxon rank sum test. Scale bar, 5 μm. **(B)** Kinetochore localization of Bub1 in nocodazole-treated cells. $G_1$-arrested *fin1Δ* cells carrying *FIN1* (pMB6) or *fin1-5A* (pMB7) plasmid and Bub1-GFP Nuf2-mCherry (3196-1-3) were released into YPD containing 20 μg/mL nocodazole for 140 min at 30˚C. White arrows indicate kinetochore localization of Fin1. The experiment was repeated three times and statistical significance determined by $p < 0.05$ using Wilcoxon rank sum test. Scale bar, 5 μm. **(C)** Accumulation of anaphase inhibitor Pds1 in *FIN1* or *fin1-5A* cells treated with nocodazole. $G_1$-arrested cells carrying *FIN1* (pMB6) or *fin1-5A* (pMB7) plasmids with Pds1-18myc (3777-2-1) were released into YPD containing 20 μg/mL nocodazole at 30˚C. Cells were collected every 20 minutes for 3 hours to prepare protein samples and count budding index. Western blot was probed with anti-myc antibody. Pgk1, loading control. The picture is representative of three experimental repeats. **(D)** *fin1-5A* cells show chromosome missegregation after release from nocodazole arrest. $G_1$-arrested cells (2167-17-2) carrying *FIN1* (pMB6) or *fin1-5A* (pMB7) plasmids were released into YPD media containing 20 μg/mL nocodazole for 120 minutes at 30˚C. After nocodazole washout, cells were collected for budding index (top left). Cells were also collected and plated onto YPD plates to examine plating efficiency, bottom left (n ≥ 300). Moreover, cells were collected over time and fixed to visualize spindle morphology (Tub1-mCherry) and *CEN*-GFP segregation. The percentage of cells with *CEN4*-GFP missegregation is shown in the top right panel (n ≥ 100). Representative images are shown in the bottom right panel. The arrow indicates *CEN4*-GFP missegregation. Scale bar, 5μm.

localization of Fin1-5A could prematurely remove Bub1 from the kinetochore, resulting in chromosome missegregation due to Bub1's roles in biorientation and checkpoint control [35,36].

## The anaphase kinetochore localization of Bub1-Bub3 is dependent on CPC activity

The phosphorylation of kinetochore proteins by Ipl1 kinase enhances SAC activity prior to anaphase onset [9,24,37]. PP1 has been shown to antagonize Ipl1 kinase during anaphase [38]. Thus, Fin1-PP1 likely antagonizes Ipl1 activity to regulate Bub1 localization at the kinetochore in anaphase. If that is the case, inhibition of Ipl1 kinase activity would abolish the anaphase

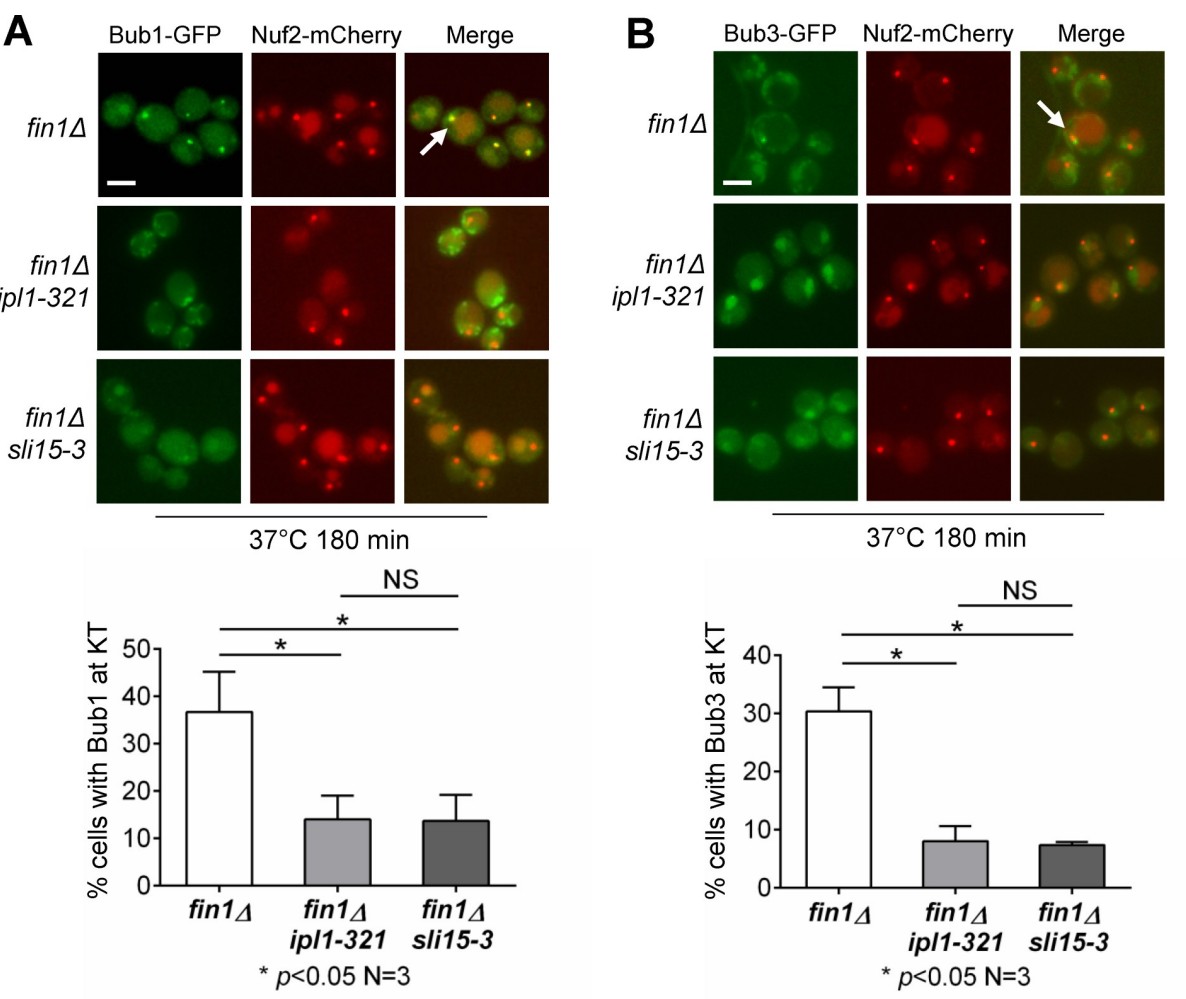

**Fig 4. Ipl1 activity is required for kinetochore localization of Bub1-Bub3 in *fin1Δ* cells during anaphase. (A)** Kinetochore localization of Bub1 in anaphase cells with impaired Ipl1 activity. Cells with the indicated genotypes and Bub1-GFP Nuf2-mCherry markers (3196-1-3, 3445-3-2, and 3653-1-1) were grown in YPD at 25°C to log-phase and then shifted to 37°C for 180 minutes to inactivate Sli15 and Ipl1. Cells were collected and counted for Bub1 kinetochore localization in anaphase cells (n = 100). White arrows indicate Bub1 anaphase kinetochore localization. The experiment was repeated three times and statistical significance determined by $p < 0.05$ using Kruskal-Wallis one-way ANOVA. Scale bar, 5 μm. **(B)** Kinetochore localization of Bub3 (3922-11-3) in anaphase cells with impaired Ipl1 activity. The same method was used to examine Bub3 kinetochore localization in *ipl1-321* (3922-2-3) and *sli15-3* (3927-9-3) mutants. The experiment was repeated three times and statistical significance determined by $p < 0.05$ using Kruskal-Wallis one-way ANOVA. Scale bar, 5 μm.

kinetochore localization of Bub1-Bub3 in *fin1Δ* cells. To test this idea, we followed Bub1 kinetochore localization in *fin1Δ*, *fin1Δ sli15-3*, and *fin1Δ ipl1-321* cells with Bub1-GFP and Nuf2-mCherry. At the restrictive temperature, the kinase activity of Ipl1 is down-regulated in both *sli15-3* and *ipl1-321* mutants, as Sli15 binds to Ipl1 and is essential for the kinase activity of Ipl1 [39]. Asynchronous cells in mid-log phase at 25°C were shifted to 37°C for 2 hours. Bub1-GFP localized to anaphase kinetochores in 37% of *fin1Δ* cells. The Bub1-GFP kinetochore localization dropped to 11% in *fin1Δ ipl1-321* and 14% in *fin1Δ sli15-3* anaphase cells (Fig 4A). To confirm that the Bub1-Bub3 complex is collectively dependent on CPC activity, we performed the same experiment using strains with Bub3-GFP. In *fin1Δ* cells, around 30% showed Bub3-kinetochore localization, however in *fin1Δ ipl1-321* and *fin1Δ sli15-3* mutant cells, the kinetochore localization was only 8% and 7%, respectively (Fig 4B). We noticed that

the frequency of kinetochore localization of either Bub1 or Bub3 in these *fin1Δ* cells was lower compared to *cdc15-2* mutants. We reason that the block of mitotic exit in *cdc15-2* mutant cells may also contribute to kinetochore localization of Bub1-Bub3 complex as described above. In any case, the data suggest that Ipl1 activity prevents the kinetochore association of Bub1 and Bub3 in anaphase. This notion appears to contradict the fact that Ipl1 localizes at the spindle during anaphase. It is likely that Ipl1 phosphorylates some kinetochore proteins before anaphase, but the reversal of this phosphorylation occurs in anaphase, which promotes the removal of Bub1-Bub3 from the kinetochore.

## Fin1-PP1 promotes the dephosphorylation of kinetochore protein Ndc80 for Bub1-Bub3 dissociation from the kinetochore in anaphase

In yeast cells, Ipl1 kinase phosphorylates a kinetochore protein Dam1 to weaken kinetochore-microtubules interaction and retain SAC activity [10,40]. Once correct chromosome attachments are established, Dam1 is dephosphorylated by PP1 to stabilize kinetochore attachment and enable SAC silencing [9]. Because Fin1-PP1 localizes to the kinetochore after SAC silencing, Dam1 is unlikely a substrate of Fin1-PP1.

Ipl1 also phosphorylates kinetochore protein Ndc80 to facilitate its microtubule interaction, and the Ipl1 phosphorylation sites in Ndc80 have been identified [41]. Evidence from mammalian cells indicates that Aurora B phosphorylates Ndc80/Hec1 to promote its binding to SAC kinase Mps1 [42]. In both yeast and mammalian cells, Mps1-dependent phosphorylation of kinetochore protein Spc105/Knl1 promotes the kinetochore binding of Bub1-Bub3 [4,43]. It is possible that Fin1-PP1 dephosphorylates Ndc80 to regulate kinetochore association of Bub1-Bub3. To test this idea, we compared Ndc80 phosphorylation in WT and *fin1Δ* cells using Phos-tag SDS-PAGE. We prepared protein samples from cells in mid-log phase, and Ndc80 showed dramatic separation under Phos-tag SDS-PAGE. We noticed an increase in highly phosphorylated species of Ndc80 in *fin1Δ* mutant cells (Fig 5A). We further measured the phosphorylation kinetics of Ndc80 in synchronous WT and *fin1Δ* cells. WT cells showed a clear reduction in the hyper-phosphorylated species of Ndc80 around anaphase entry (75 min) after $G_1$ release at 25°C. In clear contrast, *fin1Δ* mutant cells exhibited persistent appearance of hyper-phosphorylated Ndc80 species throughout the cell cycle. The ratio of phosphorylated bands to Pgk1 supports this notion (Fig 5B). These results indicate that Fin1-PP1 promotes the dephosphorylation of Ndc80.

## Ndc80 dephosphorylation decreases kinetochore localization of Bub1-Bub3 in anaphase

If Fin1-PP1 promotes the removal of SAC proteins Bub1-Bub3 by dephosphorylating Ndc80, then phospho-deficient *ndc80* mutant will likely decrease Bub1-Bub3 anaphase kinetochore localization in *fin1Δ* cells. Seven conserved Ipl1 sites (T21, S37, T54, T71, T74, S95, and S100) were identified in Ndc80, and replacement of these sites with alanine generated *ndc80-7A* mutant [24]. To confirm the decreased phosphorylation in *ndc80-7A* mutant, protein samples were prepared using asynchronous *ndc80Δ* strains covered with either *NDC80-myc* or *ndc80-7A-myc* plasmids. After separation with Phos-tag gel, a clear drop in phosphorylation was observed for the Ndc80-7A protein (Fig 6A). We also examined Ndc80 phosphorylation in *ndc80Δ fin1Δ* mutant backgrounds covered with either *NDC80-myc* or *ndc80-7A-myc* plasmids. Very little phosphorylated species were detected for Ndc80-7A protein in *fin1Δ* cells (Fig 6A), indicating that the targeting sites of Fin1-PP1 are mutated in Ndc80-7A.

To determine if the anaphase kinetochore localization of Bub1 and Bub3 depends on Ndc80 phosphorylation, we used *cdc15-2 fin1Δ ndc80Δ* strains covered with either *NDC80-myc*

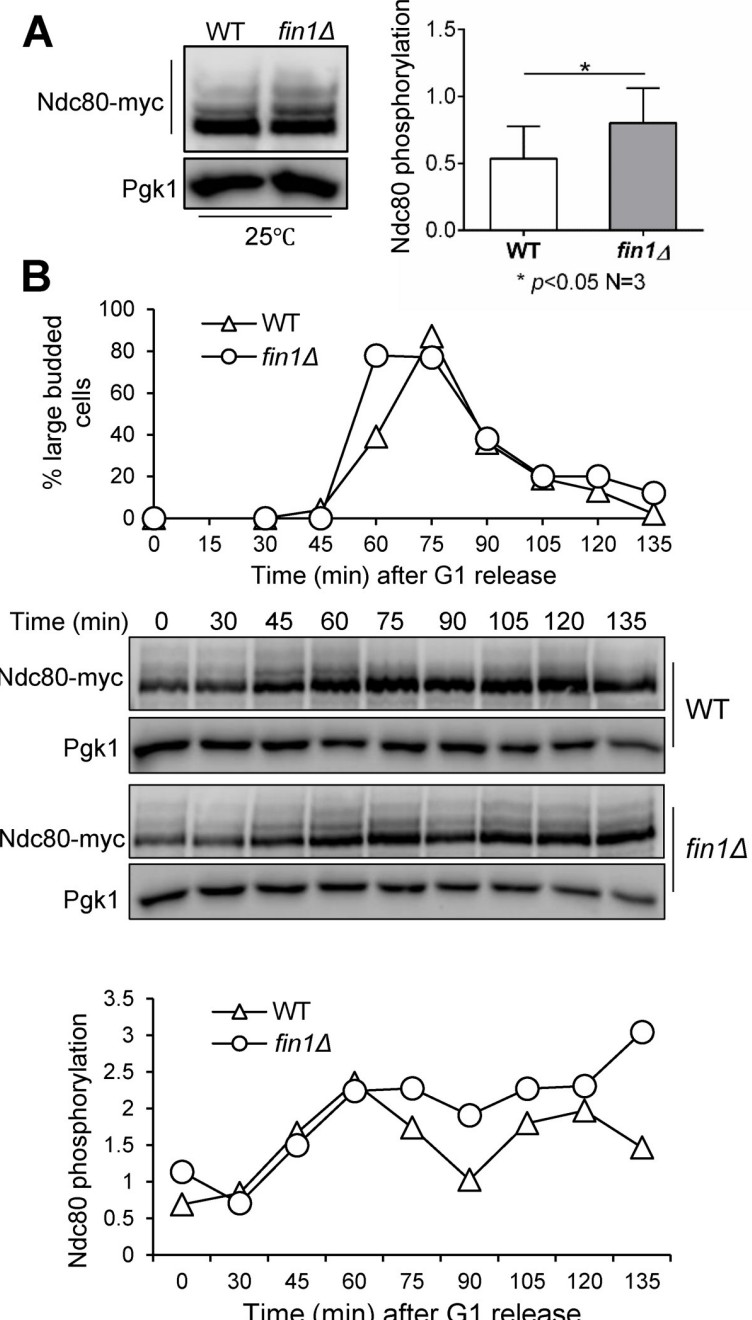

**Fig 5. Fin1-PP1 promotes dephosphorylation of Ipl1 substrate Ndc80 in anaphase. (A)** Analysis of Ndc80 phosphorylation in WT (3802-1-4) and *fin1Δ* (3802-2-2) cells using Phos-tag gel. WT and *fin1Δ* cells with Ndc80-13myc were grown to log phase in YPD at 30°C. Protein samples were collected and separated on Phos-tag gel. The membrane was probed with anti-myc antibody to show separation of phosphorylated Ndc80 species. The image shown is a representative of three experimental repeats and statistical significance determined by $p < 0.05$ using Wilcoxon rank sum test. **(B)** *fin1Δ* cells show delayed Ndc80 dephosphorylation. WT (3802-1-4) and *fin1Δ* (3802-2-2) cells containing Ndc80-13myc were arrested in $G_1$ and then released into YPD at 30°C. Cells were collected every 15 minutes to prepare protein samples and count budding index. Phos-tag gel was used for protein separation and show phosphorylated Ndc80 protein species. The ratio of phosphorylated species to Pgk1 loading control was quantified using ImageJ for each time point and graphed. Pictures are representative from three experimental repeats.

or *ndc80-7A-myc*. The cells were grown to log phase at 25˚C and then shifted to the non-permissive temperature of 37˚C for 3 hours to arrest cells in telophase. The signals of Bub1-GFP and Nuf2-mCherry were examined, and we found that Bub1 localized to the kinetochore in 47% cells covered with *NDC80*, while Bub1 anaphase kinetochore localization was detected in 22% cells covered with *ndc80-7A* (Fig 6B). Similar results hold true for Bub3; 55% of cells with *NDC80* showed Bub3-GFP anaphase kinetochore localization, but only 19% of cells covered with *ndc80-7A* exhibited Bub3 kinetochore localization (Fig 6C). It is worth noting that *cdc15-2 ndc80Δ* cells covered with *NDC80-myc* plasmids show decreased Bub1 and Bub3 anaphase kinetochore localization compared to *cdc15-2* cells (Fig 2). One possibility is that the myc-tag at the C-terminus of Ndc80 compromises its function. We also recognize that the kinetochore localization of Bub1 and Bub3 is not completely abolished in *ndc80-7A* cells, indicating that Fin1-PP1 may also dephosphorylate other sites in Ndc80 or other substrates, such as Bub1, for the kinetochore dissociation of Bub1-Bub3. Regardless, the decreased anaphase kinetochore localization of Bub1 and Bub3 in cells expressing *ndc80-7A* suggests that Ndc80 dephosphorylation contributes to Bub1 and Bub3 removal from the kinetochore during anaphase.

*fin1Δ* cells show persistent Ndc80 phosphorylation and Bub1-Bub3 localization at the kinetochore. This might be a result of lingering Mps1 kinase at the kinetochore. To test this, we performed a kinetochore-immunoprecipitation assay in *cdc15-2* and *cdc15-2 fin1Δ* cells expressing Dsn1-FLAG and Mps1-13myc as described previously [44]. No increase in Mps1 interaction with kinetochore protein Dsn1 was detected in *cdc15-2 fin1Δ* cells compared to *cdc15-2* cells (S6 Fig). Additionally, we examined if the anaphase kinetochore localization of Bub1-GFP in *fin1Δ* cells was dependent on Mps1 activity using a temperature-sensitive *mps1-1* mutant. Interestingly, we found that Bub1 co-localization with the kinetochore in *mps1-1 fin1Δ* cells was greatly reduced compared to *fin1Δ* cells (S7 Fig). Collectively, these results suggest that anaphase kinetochore localization of Bub1 in *fin1Δ* cells is dependent on Mps1 activity. We speculate that Ndc80 dephosphorylation by Fin1-PP1 may downregulate Mps1 activity or upregulate the dephosphorylation of Mps1 substrates to facilitate Bub1-Bub3 removal from the kinetochore, but this unlikely occurs through kinetochore localization of Mps1.

## Phospho-deficient *ndc80-7A* mutants show viability loss in cells with induced syntelic attachment

Premature Fin1-PP1 localization to kinetochores causes a viability loss in cells with induced syntelic attachment [19]. We reasoned that untimely dephosphorylation of Ndc80 by Fin1-PP1 may have a similar phenotype in response to syntelic attachment. We previously showed that overexpression of the coiled-coiled domain of Cik1 motor protein (*CIK1-CC)* induces syntelic attachments [7]. Indeed, phospho-deficient *ndc80-7A* mutants grew poorly on galactose plates, where *CIK1-CC* is overexpressed to induce syntelic attachment (Fig 7A). In addition, *ndc80-7A* cells displayed significant viability loss following *CIK1-CC* overexpression. Only 33% of *ndc80-7A* mutants were viable after *CIK1-CC* overexpression for 6 hours, whereas 77% of WT cells were viable under the same condition (Fig 7B).

If the kinetochore localization of Fin1-PP1 is upstream of Ndc80 dephosphorylation, then phospho-deficient *ndc80-7A* will be sensitive to incorrect chromosome attachments even in *fin1Δ* mutant cells. As expected, *fin1Δ ndc80-7A* cells were also sensitive to *CIK1-CC* overexpression that induces syntelic attachments, as indicated by the poor growth on galactose plates (Fig 7C). *fin1Δ ndc80-7A* cells lost viability after *CIK1-CC* overexpression for 6 hours with only 24% viable cells compared to 26% in *ndc80-7A* single mutant (Fig 7D). Together, these

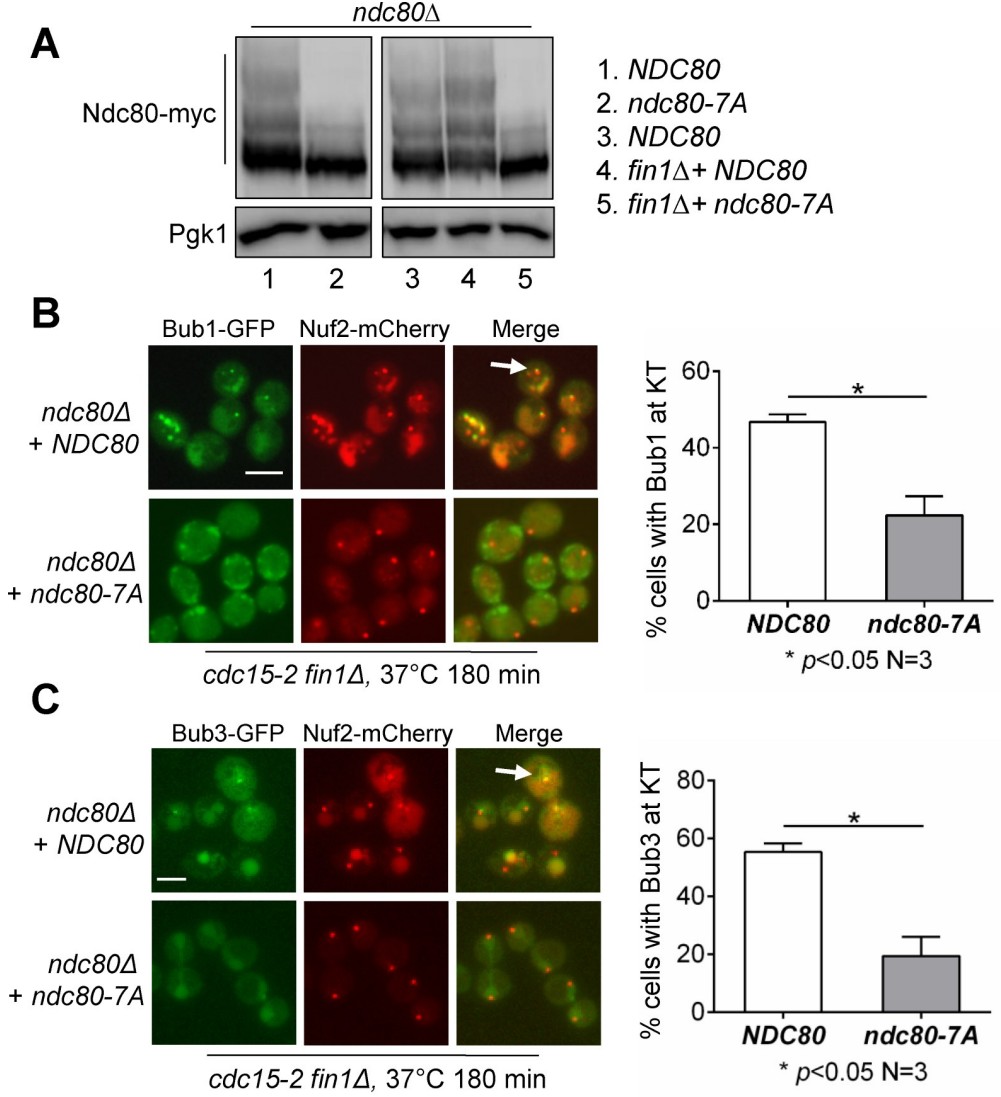

**Fig 6. Ndc80 dephosphorylation promotes Bub1 dissociation from the kinetochore during anaphase. (A)** Reduced phosphorylation for phospho-deficient Ndc80-7A proteins. *fin1Δ ndc80Δ* cells carrying WT *NDC80-myc* (3893-10-3) and phospho-deficient *ndc80-7A-myc* (3897-5-3) plasmids were grown to log phase at 30°C. Protein samples were prepared from asynchronous cells and run on Phos-tag gel. Western blot was probed with anti-myc antibody to show separation of phosphorylated species. **(B)** Bub1 kinetochore localization in *ndc80-7A* cells during anaphase. *fin1Δ ndc80Δ cdc15-2 BUB1-GFP NUF2-mCherry* cells carrying either *NDC80* (3975-5-3) or *ndc80-7A* (3941-1-4) plasmids were grown to log phase in YPD at 30°C then shifted to the non-permissive temperature of 37° for 180 minutes to inactivate Cdc15. Samples were collected and pictures taken. Anaphase cells with Bub1-kinetochore co-localization was counted (n = 100). White arrows indicate co-localization. The experiment was repeated three times and statistical significance determined by *p* < 0.05 using Wilcoxon rank sum test. Scale bar, 5 μm. **(C)** Bub3 kinetochore localization in *ndc80-7A* cells during anaphase. The same protocol was used to examine kinetochore localization of Bub3 (3893-9-2 and 4057-26-1). The experiment was repeated three times and statistical significance determined by *p* < 0.05 using Wilcoxon rank sum test. Scale bar, 5 μm.

results suggest that Ndc80 dephosphorylation causes viability loss when syntelic chromosome attachment occurs. We speculate that decreased kinetochore binding of Bub1-Bub3 in *ndc80-7A* cells results in premature SAC silencing and/or increased erroneous kinetochore attachments, leading to viability loss when syntelic attachment is induced.

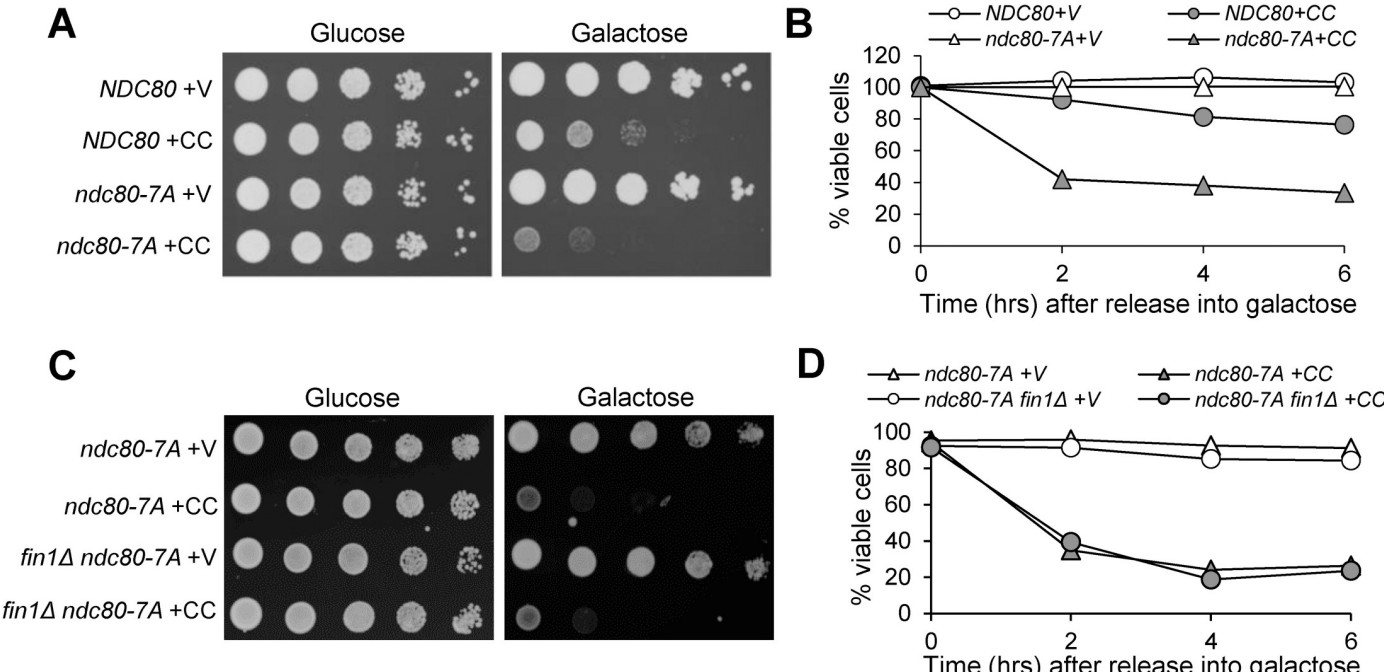

**Fig 7. Phospho-deficient *ndc80-7A* mutant is sensitive to the induction of syntelic chromosome attachments. (A)** *ndc80-7A* mutant cells are sensitive to *CIK1-CC* overexpression. *NDC80* (SBY7258) and *ndc80-7A* (SBY7259) cells carrying vector (V, pRS416) or $P_{GAL}CIK1$-*CC* (CC, pYW200) plasmid were grown to saturation, 10-fold serially diluted and plated onto glucose and galactose plates. The plates were incubated at 30˚C for 2 days before scanning. **(B)** The same strains were grown to log-phase in raffinose media, then galactose was added to a final concentration of 2%. The cells were collected every 2 hours and spread onto YPD plates. Plates were grown at 30˚C overnight for plating efficiency (n $\geq$ 300). **(C)** *fin1Δ* mutant does not suppress the sensitivity of *ndc80-7A* cells to *CIK1-CC* overexpression. *ndc80-7A* (3932-5-1) and *fin1Δ ndc80-7A* (3932-3-1) cells carrying vector (V, pRS416) or $P_{GAL}CIK1$-*CC* (CC, pYW200) plasmid were grown to saturation, 10-fold serially diluted and plated onto glucose and galactose plates. Plates were grown at 30˚C for 2 days before scanning. **(D)** The same strains were used to determine the viability as described above (n $\geq$ 300).

## Discussion

Mistakes in chromosome attachment activate the SAC to prevent anaphase onset. The establishment of bipolar attachment on all chromosomes silences the SAC to allow anaphase progression. A brief release of Cdc14 phosphatase from the nucleolus occurs in early anaphase via activation of the FEAR pathway. The dephosphorylation of the CDK substrate Fin1 by Cdc14 triggers the recruitment of Fin1-PP1 complex to the kinetochore [22]. We previously showed that the kinetochore recruitment of Fin1-PP1 removes Bub1 from the kinetochore during anaphase [19]. In this work, we further found that the kinetochore recruitment of Fin1-PP1 prevents kinetochore association of both Bub1 and Bub3, but does not affect Mad1. In addition, we showed that Fin1 was required for efficient dephosphorylation of kinetochore protein Ndc80, an Ipl1 kinase substrate. Moreover, a phospho-deficient *ndc80* mutant reduced Bub1--Bub3 kinetochore localization in anaphase in *fin1Δ* cells and showed a dramatic viability loss in response to tensionless syntelic attachment. Our results support the conclusion that kinetochore recruitment of Fin1-PP1 during anaphase leads to Ndc80 dephosphorylation, which compromises the kinetochore association of Bub1-Bub3 SAC complex. Therefore, this research uncovered a molecular mechanism that controls kinetochore localization of SAC proteins Bub1-Bub3 during anaphase.

One open question is how Ndc80 phosphorylation regulates kinetochore localization of Bub1 and Bub3. The SAC kinase Mps1 binds to the N-terminal tail of Ndc80 and phosphorylates kinetochore protein Spc105/Knl1, which subsequently recruits Bub1-Bub3 to the

kinetochore [4,41,45,46]. After Bub1 binds to Spc105, it is also phosphorylated by Mps1 to promote the further recruitment of Mad1-Mad2, resulting in SAC activation [43,47]. How Mps1 is recruited to Ndc80 is not entirely understood. Previous data suggest that Ipl1 phosphorylates the T676 residue within Mps1 to allow a conformational change that facilitates binding to Ndc80, competing with Ndc80-microtubule interactions [46,48,49]. However, other reports in mammalian cells indicate that Aurora B-dependent phosphorylation of Ndc80 controls its binding to Mps1 in multiple ways. Ndc80 N-terminal tail phosphorylation by Aurora B decreases Ndc80's affinity for microtubules, while increasing its affinity for Mps1, creating a positive feedback loop on Mps1 recruitment and SAC activation [42,50]. We used co-IP method to assess kinetochore binding of Mps1 in *cdc15-2* and *cdc15-2 fin1Δ* cells, but no increased Mps1 kinetochore binding was observed in *cdc15-2 fin1Δ* cells compared to *cdc15-2*. We also examined Spc105 phosphorylation, which is dependent on Mps1, but we could not detect any difference in WT and *fin1Δ* cells using Phos-tag method. Therefore, Fin1-PP1 likely regulates Bub1-Bub3 kinetochore localization through a manner independent of Mps1 kinetochore localization.

Although strong kinetochore association of Bub1-Bub3 complex was noticed in *cdc15-2 fin1Δ* mutant cells in anaphase, no Mad1 was detected at the anaphase kinetochore in *fin1Δ* cells (S2 Fig). It is likely that distinct mechanisms control the removal of Mad1-Mad2 from the kinetochore for SAC silencing and the removal of Bub1-Bub3 during anaphase [47,51]. For example, kinetochore protein Spc105 associates with PP1, which might contribute to Mad1--Mad2 dissociation from the kinetochore for SAC silencing [52]. Another possibility is that motor protein Cin8 recruits PP1 to the kinetochore for SAC silencing [53,54].

In this work, we found that Fin1-PP1 promotes Ndc80 dephosphorylation. It is possible that kinetochore recruitment of Fin1-PP1 reverses Ndc80 phosphorylation for kinetochore removal of Bub1-Bub3. Because recent evidence indicates that Fin1 is required for the assembly of normal levels of Dam1 and Ndc80 kinetochore submodules [55], Fin1-dependent dephosphorylation might promote this assembly. Moreover, this assembly may indirectly facilitate Bub1-Bub3 removal from the kinetochore. Our data showed the Bub1-Bub3 removal in phospho-deficient *ndc80-7A* mutant is only partial (Fig 6), indicating that either additional Ipl1 sites present in Ndc80-7A or other mechanisms may also play a role in removing Bub1--Bub3 from the kinetochore. For example, we detected delayed dephosphorylation of Bub1 in *fin1Δ* cells [19], thus this dephosphorylation could also contribute to kinetochore delocalization of Bub1-Bub3 during anaphase.

Anaphase onset occurs after SAC silencing. Why do cells need an extra mechanism to remove SAC proteins during anaphase after SAC silencing? One hypothesis is to prevent SAC reactivation during anaphase when kinetochore attachment becomes tensionless due to resolution of cohesion between sister chromatids [56,57]. Moreover, the kinetochore retention of SAC proteins may also enable SAC activation to delay anaphase progression under some physiological conditions, such as cell stress caused by DNA damage. We have shown that DNA damage prevents the activation of FEAR and the subsequent Cdc14 release from nucleolus [58]. Bmh1 binds to phosphorylated Fin1 to inhibit kinetochore association of Fin1-PP1, and *bmh1* mutant cells are sensitive to DNA damage [59]. Therefore, more work is needed to examine if the kinetochore association of Bub1-Bub3 is required for checkpoint reengagement when DNA damage is induced during anaphase. A recent study in mammalian cells shows that Bub1-Bub3 complex promotes DNA synthesis at telomere regions [60]. Thus, another possibility is that the temporal release of Bub1-Bub3 from kinetochores might be critical for the timing control for DNA replication at different regions of chromosomes.

Our results from budding yeast suggest that the removal of SAC proteins Bub1 and Bub3 involves kinetochore localization of Fin1-PP1 and dephosphorylation of kinetochore protein

Ndc80 by Fin1-PP1. This removal is regulated by Cdc14 phosphatase that reverses CDK-dependent phosphorylation of Fin1 during anaphase, and this regulation might be conserved from yeast to human cells. In both yeast and mammalian cells, the CPC kinase Ipl1/Aurora B phosphorylates Ndc80/Hec1 to destabilize kinetochore-microtubule interaction and enhance Mps1 recruitment to kinetochores [24,42]. In addition, in mammalian cells CDK activity prevents the chromatin/kinetochore localization of a PP1 regulator, Repo-man, prior to anaphase onset [61]. Following anaphase entry, Repo-Man-PP1 localizes to the chromatin/kinetochore to antagonize Aurora B activity, which stabilizes the kinetochore-microtubule interaction [62]. Thus, the Repo-Man-PP1 complex could be a functional orthologue of yeast Fin1-PP1. Collectively, our results reveal an orchestrated effort by Cdc14 and Fin1 to remove SAC proteins from the kinetochore following anaphase onset by targeting PP1 to the kinetochore. Therefore, this regulation shifts kinase-phosphatase balance at kinetochores to fully remove SAC proteins, which may prevent SAC reengagement during anaphase. The temporal control of this regulation is critical for faithful chromosome segregation.

## Materials and methods

### Yeast strains, growth, and media

The relevant genotypes and sources of the yeast strains used in this study are listed in S1 Table. All the strains listed are isogenic to Y300, a W303 derivative, and they were constructed by tetrad dissection. PCR-based method was used to construct *BUB3-GFP* and *MPS1-13myc* strains [63]. Yeast cell growth, synchronization, and *CIK1-CC* overexpression were performed as described previously [7]. The plasmids used in this study are listed in S2 Table.

### Cytological techniques

For fluorescence microscopy, collected yeast cells were fixed with 3.7% formaldehyde for 5 minutes and then washed once with water. Cells were then re-suspended in $1 \times$ PBS (pH 7.2) for the examination of fluorescence signals using a microscope with a 60× objective (BZ-X800 from Keyence). Cells with Bub1-GFP, Bub3-GFP, Mad1-GFP and Nuf2-mCherry were collected and imaged in the same manner without fixation due to low GFP signal. Live-cell microscopy was carried out with the Andor Revolution Spinning Disk imaging system. Cells were spotted onto an agarose pad filled with synthetic complete medium and 10% glucose. All live-cell images were acquired at 25°C with a 100× objective lens. A Z-stack with 13 planes of 0.4μm was acquired every 3 minutes and converted to a maximum projection using Andor IQ2 software.

### Western blotting

Yeast cells (1ml) were collected by centrifugation and the cell pellets were re-suspended in 200μl of 0.1 M NaOH. After incubation at room temperature for 5 minutes, the sample was centrifuged, and the pellets were re-suspended in 100μl $1 \times$ SDS protein loading buffer. The protein samples were boiled for 5 minutes and resolved by 8% SDS-PAGE. After probing for anti-myc antibodies (BioLegend) followed by horseradish peroxidase (HRP)-conjugated secondary antibody (Cell Signaling Technology), the proteins were detected with enhanced chemiluminescence (ECL, PerkinElmer). Bio-Rad ChemiDoc imaging system was used to image blots.

Ndc80 phosphorylation was detected using Phos-tag SDS-PAGE. We added 20μM Phos-tag and 150μM $MnCl_2$ to SDS-PAGE gel which was run at 20mA/gel for 120 minutes in the cold. The gel was washed with transfer buffer containing 10mM EDTA for 10 minutes and

then washed with transfer buffer alone for 10 minutes. The blots were then prepared as described above.

## Co-immunoprecipitation of Mps1 and Dsn1

Cells with the indicated genotypes (15 mL) were grown in YPD media overnight at 25°C to log phase. Cells were then switched to 37°C for two hours to inactive Cdc15-2 and harvested at 4,000 rpm (Eppendorf Centrifuge 5810R) for 1 minute. Cells were washed once with water and resuspended in RIPA buffer (50 mM Tris-HCl pH 7.5, 150 mM NaCl, 5 mM EDTA, 0.05% Tween-20) along with sodium azide, MG-132 inhibitor, PMSF, and protease inhibitor cocktail (EMD Millipore Corp., Billerica, MA). Samples were frozen in liquid nitrogen and subsequently crushed using a CyroMill. Samples were thawed on ice in the presence of MG-132 inhibitor. After centrifugation at 3,000 g for 20 minutes at 4°C, the supernatant was collected and centrifuged again at 20,000 g for 20 minutes at 4°C. After input samples were taken, M2 FLAG agarose beads (Sigma Aldrich) were added to immunoprecipitate Dsn1-FLAG overnight at 4°C on a rotor. Beads were washed three times, resuspended in 1x SDS loading buffer, and incubated at 65°C for 10 minutes. Western blotting was performed using anti-FLAG, anti-myc, and anti-Pgk1 antibodies.

## Statistical analysis

To assess any difference in protein levels and phosphorylation among yeast strains, we used ImageJ to quantify the intensity of bands on western blot images. For Ndc80 phosphorylation, a ratio was determined by the intensity of the phosphorylated species over the intensity of Pgk1 band for each lane. The difference in phosphorylation was analyzed using Wilcoxon rank sum test to determine $p$ values. Statistical significance was determined when $p < 0.05$ (*).

Results from fluorescence microscopy experiments was determined by counting 100 cells for each relevant yeast strain. We repeated each experiment a total of three times. We then performed either a non-parametric Wilcoxon rank sum test to compare WT to one mutant, or a non-parametric Kruskal-Wallis one-way ANOVA to compare WT to two mutants; and determine $p$ values. Statistical significance was determined when $p < 0.05$ (*) and is denoted as such.

## Supporting information

**S1 Table. The list of yeast strains used in this study.**
(DOCX)

**S2 Table. The list of plasmids used in this study.**
(DOCX)

**S1 Fig. Fin1 promotes Bub3 dissociation from the kinetochore during anaphase.** (A) Live-cell imaging of WT (4065-8-1) and fin1Δ (4065-5-1) cells to show kinetochore association of Bub3 in anaphase. Log-phase WT and fin1Δ cells with Bub3-GFP and Nuf2-mCherry were spotted onto an agarose pad filled with synthetic complete medium and subjected to live-cell microscopy. We set time 0 as the last point when the distance between two Nuf2-mCherry foci is less than 3μm. Arrows indicate co-localization of Bub3-GFP and Nuf2-mCherry. (B) Cells that had Bub3-GFP co-localization with Nuf2-mCherry at any time during anaphase was counted as positive and the percentage of positive cells was graphed (n = 20). KT: kinetochore. (C) The co-localization of Bub3-GFP with the kinetochore is dynamic after anaphase onset. The co-localization of Bub3 with either one or both kinetochore clusters during anaphase was counted for each frame of WT and fin1Δ cells during anaphase. We counted 10 cells for each

strain.
(TIF)

**S2 Fig. Mad1 kinetochore localization during anaphase.** (A) cdc15-2 (3460-1-2) and cdc15-2 fin1Δ (3457-1-3) cells carrying Mad1-GFP and Nuf2-mCherry were grown to log phase at 25˚C and then shifted to 37˚C for 180 minutes. Samples were collected and pictures were taken. The frequency of cells without co-localization of Mad1-GFP and Nuf2-mCherry signals was counted for each strain (n = 100). The picture is representative of three experimental repeats and statistical significance determined by p < 0.05 using Wilcoxon rank sum test. Scale bar, 5 μm. (B) WT (3460-1-2) and fin1Δ (3457-1-3) cells containing Mad1-GFP and Nuf2-mCherry were grown to log phase at 25˚C. Asynchronous cell samples were collected and pictures were taken. The frequency of co-localization of Mad1-GFP and Nuf2-mCherry signals during anaphase was counted for each strain (n = 100). Anaphase was determined when the distance between the two Nuf2 foci was greater than 3μM. The picture is a representative of three experimental repeats and statistical significance determined by p < 0.05 using Wilcoxon rank sum test. Scale bar, 5 μm.
(TIF)

**S3 Fig. The kinetochore localization of Bub1 and Bub3 in anaphase is interdependent.** (A) WT (2827-1-4) and bub3Δ (3887-3-4) cells containing Bub1-GFP and Nuf2-mCherry were grown to log phase at 25˚C. Asynchronous cells were collected and pictures were taken. The frequency of co-localization of Bub1-GFP and Nuf2-mCherry signals in metaphase was counted for each strain (n = 100). Metaphase cells are large-budded with two Nuf2 clusters less than 3μm apart. White arrows indicate Bub1-kinetochore co-localization. The experiment was repeated three times and statistical significance was determined by p < 0.05 using Wilcoxon rank sum test. Scale bar, 5 μm. (B) WT (4065-8-1) and bub1Δ (4065-3-1) cells containing Bub3-GFP and Nuf2-mCherry were grown as described above. The same method was used to compare the frequency of Bub3-Nuf2 co-localization. Scale bar, 5 μm.
(TIF)

**S4 Fig. The localization of Fin1-GFP and phospho-deficient Fin1-5A-GFP in cdc14-2 mutant cells.** cdc14-2 (3536-1-2) cells with FIN1-GFP (pSB1252) or fin1-5A-GFP (pSB1359) plasmid were grown at 25˚C until log-phase, and then transferred to 37˚C for 180 minutes. Cells were collected and fixed to visualize GFP signal. Cells with different patterns of GFP signal were counted (n = 100). Representative images are shown on the top. Scale bar; 5μm. The percentage of cells with different patterns of GFP signals is shown in the bottom panel.
(TIF)

**S5 Fig. The SAC is intact in cells expressing Fin1-5A.** (A) ctf13-30 kinetochore mutant cells expressing Fin1-5A show metaphase arrest after long time incubation at non-permissive temperature. G1-arrested WT (4159-6-4) and ctf13-30 (4159-4-1) cells carrying FIN1 (pMB6) or fin1-5A (pMB7) plasmid and Tub1-GFP were released into 37˚C YPD media for 5 hours. Cells were collected and fixed every 30 minutes to visualize spindle morphology (Tub1-GFP). The percentage of cells with short spindles was counted for each time point (n = 100). Spindles less than 3μM in length were counted as short spindles. Representative images were taken at 5 hours. Scale bar, 5μM. (B) Cells expressing Fin1-5A do not show increased chromosome mis-segregation. G1-arrested CEN4-GFP Tub1-mCherry cells (2167-17-2) carrying FIN1 (pMB6) or fin1-5A (pMB7) plasmid and were released into YPD media at 30˚C. Cells were collected over time for budding index (top left). Cells at 80 minutes were plated onto YPD plates to examine plating efficiency (n ≥ 300) (Bottom left). We also collected cells at 80 minutes and fixed them to visualize spindle morphology (Tub1-mCherry) and CEN4-GFP segregation. The

percentage of cells with CEN4-GFP missegregation is shown in the top right panel (n ≥ 100). Representative images are shown in the right bottom panel. Scale bar, 5μm.
(TIF)

**S6 Fig. The absence of Fin1 does not increase kinetochore binding of Mps1.** cdc15-2 MPS1-13myc (DS002), cdc15-2 fin1Δ MPS1-13myc (DS003), cdc15-2 MPS1-13myc DSN1-FLAG (DS004), and cdc15-2 fin1Δ MPS1-13myc DSN1-FLAG (DS005) cells were grown overnight in 15 mL YPD media at 25°C then switched to 37°C for 2 hours to inactive Cdc15-2. Cells were harvested and lysed via CryoMill, and Dsn1-FLAG protein was immunoprecipitated using M2 anti-FLAG beads. Dsn1-FLAG and Mps1-13myc were detected using anti-FLAG and anti-myc antibodies after separation on a 10% SDS-PAGE gel. Pgk1, loading control. * a non-specific band.
(TIF)

**S7 Fig. Anaphase kinetochore localization of Bub1 in fin1Δ cells is dependent on Mps1 kinase activity.** WT (3887-4-1), fin1Δ (3196-1-3), mps1-1 (4146-2-4), and mps1-1 fin1Δ (4171-3-1) cells containing Bub1-GFP and Nuf2-mCherry were grown to log phase at 25°C then switched to 37°C for 150 minutes. The frequency of co-localization of Bub1 and Nuf2 was counted in metaphase and anaphase cells (top right). WT and fin1Δ cell in anaphase were determined when the distance between two Nuf2 foci was greater than 3μm (n = 100), and the number is shown in the bottom right panel. No elongated spindles were observed in mps1-1 mutant cells after incubation at 37°C because of the function of Mps1 in spindle pole body duplication. The co-localization of Bub1-GFP with the single Nuf2 cluster was counted in mps1-1 mutant cells (n = 100), and the number is shown in the bottom right panel. The experiment was repeated three times and statistical significance of metaphase localization was determined by $p < 0.05$ using Wilcoxon rank sum test. Statistical significance of anaphase localization was determined by $p < 0.05$ using Kruskal-Wallis one-way ANOVA. Scale bar, 5 μm.
(TIF)

## Acknowledgments

We are grateful to the yeast community at Florida State University for reagents and helpful suggestions. We thank Drs. Sue Biggins, Georjana Barnes, and Frank Uhlmann for providing yeast strains and plasmids. We thank Ruth Didier, Ernest Philips and Dr. Akash Gunjan for their assistance with live-cell imaging.

## Author Contributions

**Conceptualization:** Michael Bokros, Delaney Sherwin, Yanchang Wang.

**Data curation:** Delaney Sherwin.

**Formal analysis:** Michael Bokros, Delaney Sherwin, Yanchang Wang.

**Funding acquisition:** Yanchang Wang.

**Investigation:** Michael Bokros, Delaney Sherwin, Marie-Helene Kabbaj, Yanchang Wang.

**Methodology:** Michael Bokros, Delaney Sherwin, Marie-Helene Kabbaj.

**Project administration:** Yanchang Wang.

**Resources:** Yanchang Wang.

**Supervision:** Yanchang Wang.

**Writing – original draft:** Michael Bokros, Delaney Sherwin.

**Writing – review & editing:** Yanchang Wang.

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
