## [Decision Letter · Decision Letter 0]

4 Jan 2021

Dear Dr Wang,

Thank you very much for submitting your Research Article entitled 'The mechanism that promotes kinetochore dissociation of spindle assembly checkpoint proteins Bub1 and Bub3 in anaphase' to PLOS Genetics.

The manuscript was fully evaluated at the editorial level and by independent peer reviewers. The reviewers appreciated the attention to an important problem, but raised some substantial concerns about the current manuscript. Based on the reviews, we will not be able to accept this version of the manuscript, but we would be willing to review a revised version. We cannot, of course, promise publication at that time.

In particular the reviewers expressed enthusiasm for the insights into Bub1/3 regulation, but indicated that the central conclusions need further support, and that some of the claims need to be tempered (this is particularly true for Reviewer #2).

If you decide to revise the manuscript for further consideration at PLOS Genetics, please aim to resubmit within the next 60 days, unless it will take extra time to address the concerns of the reviewers, in which case we would appreciate an expected resubmission date by email to plosgenetics@plos.org.

[LINK]

We are sorry that we cannot be more positive about your manuscript at this stage. Please do not hesitate to contact us if you have any concerns or questions.

Yours sincerely,

Gregory P. Copenhaver

Editor-in-Chief

PLOS Genetics

Reviewer's Responses to Questions

**Comments to the Authors:**

Reviewer #1: This is an interesting manuscript that sheds new light onto the removal of spindle checkpoint (SAC) proteins from kinetochores in anaphase yeast cells. The major conclusion is that Fin1 (presumably as part of the protein phosphatase 1 complex) dephosphorylates Ndc80 at the kinetochore, which in turn prevents localization of the Bub1-Bub3 complex at kinetochores. This seems to be a key step in the mechanism that keeps the SAC off in anaphase. The experiments have been designed logically and the data are largely compelling. This should be an impactful paper. There are some minor issues and comments that can be addressed.

The authors first build on a previous observation that Fin1 prevents Bub1 kinetochore association in anaphase, by using live cell microscopy to demonstrate that Bub1 dynamically re-associates with kinetochores in the fin1 null mutant. They go on to show that Bub1’s partner, Bub3, also inappropriately associates with kinetochores and that this likely requires a Bub1-Bub3 complex. This co-dependence also held true in nocodazole arrested cells, which is intriguing. This is all interesting new information, not just about SAC silencing, but also about the SAC activation mechanism.

They used a fin1-5a mutant that prematurely associates with kinetochores (before anaphase and in nocodazole arrest) to show that its localization is not sufficient for displacing the Bub1-Bub3 complex. Several metrics were used to show that Fin1-5A at kinetochores does not inactivate the SAC, thought it was not ruled out that the SAC might be weakened. Longer time courses would be needed for that. It might be worth pointing this out in the text.

After a nocodazole release, fin1-5A cells had reduced viability, but this should be done relative to plating untreated cells (no nocodazole). The experiments following CEN4 segregation are interesting, showing mis-segregation in fin1-5A cells after nocodazole release. If untreated fin-5A cells have reduced viability, then it might be worth following CEN4 segregation in untreated cells (I don’t think this is a necessary experiment for this manuscript, but it would be interesting to follow up).

Very interestingly, they present evidence that Bub1 kinetochore localization in anaphase depends on the CPC. The CPC should be relocalized to the spindle midzone in these anaphase cells, so this suggests CPC is staying at kinetochores. This would be a very valuable follow up study (but again, not necessary for this manuscript).

Evidence is presented that the relevant substrate of Fin1 is Ndc80, which has a persistent hyper-phosphorylated form in fin1 null cells. Using an Ndc80 mutant lacking seven phosphorylation sites (ndc80-7A), they provide good evidence that persistent Ndc80 phosphorylation in fin1 null cells at least partly accounts for the retained Bub1-Bub3 at kinetochores. In agreement, ndc80-7A was sensitive to syntelic kinetochore attachments. This provides important biological context.

They speculate that Mad1 dissociation from kinetochores occurs through a different mechanism, since Mad1 is not at the Bub1-Bub3 positive kinetochores in fin1 null telophase cells. This is interesting. It would be valuable to look in anaphase cells to be sure this result holds true (presently the experiment was done in the telophase arrested cdc15-2 background). This would be best investigated in a follow up study. There is already a lot of data in the current manuscript, and although it will be very interesting to follow up the Mad1 experiments, this is likely to open a whole lot of new questions.

Reviewer #2: Major conclusion: Fin1-PP1 counteracts Ipl1 phosphorylation at KT to negatively regulate Bub3/Bub1 localization in anaphase.

Summary:

Bokros and colleagues address the role of Fin1 in regulating anaphase localization of Bub3 and Bub1 checkpoint proteins in yeast. This is a very interesting area because while the spindle assembly checkpoint (SAC) prevents anaphase onset, once the checkpoint is satisfied and chromosome segregation ensues, the low-tension kinetochore-microtubule attachments in anaphase don’t re-engage the checkpoint. This is broadly thought to be due to phosphatases counteracting kinase activity, though the precise mechanisms silencing the checkpoint and preventing reactivation are still unknown. Here the authors show the PP1 phosphatase regulator Fin1 is required for the removal of checkpoint proteins Bub3 and Bub1 in anaphase. Prematurely localized Fin1 at the kinetochore contributes to defects in cells responding to syntelic attachments, suggesting an antagonistic relationship with the Ipl1 kinase that promotes turnover of these incorrect attachments. The authors then go on to show that Ipl1 is required for the fin1 defects observed in cells, and that this is partially mediated through Ipl1 phosphorylation of the Ndc80 kinetochore protein.

Taken together, these results should be of interest to the field as they help broaden our mechanistic understanding of the regulation of checkpoint silencing, although the authors point out that Mad1 regulation is still intact so additional key regulation needs to be discovered. However, the major concern with this manuscript is that many interpretations put forth by the authors are not directly supported by the experimental data. These concerns need to be addressed to make the conclusions in this manuscript.

Major Points:

1) Spc105 appears to be the major binding site for PP1 at KT (Rosenberg 2011), where Glc7/PP1 can then remove Mps1 phosphorylation at MELT motifs which are responsible for recruiting Bub3/Bub1. It is possible that the ipl1 (as well as ndc80-7A) and fin1 associated phenotypes are mediated through Mps1, so the authors need to analyze Mps1 localization. To address whether Mps1 is still at the KT and contributing to persistent Bub3/Bub1 anaphase localization, the authors should perform ChIP or kinetochore IPs (or any technique that will work to detect Mps1) from anaphase lysates in WT and fin1 mutants. Either result from such an IP (no change in Mps1 loading or Fin1 dependent change) would greatly strengthen (no change) or weaken (Fin1 dependent change) their model and thus should be addressed. Alternatively, the authors could use mps1 ts mutants to test if the Bub3/Bub1 anaphase localization in fin1 cells is independent of Mps1.

2) The claim that Ipl1 regulates Bub1/3 localization in anaphase is fine (but a bit odd considering Ipl1 is not on anaphase kinetochores), but the authors do not address the possibility that the ipl1 phenotype is due to prematurely localized PP1 which then leads to loss of Bub1/3 localization.

3) In multiple cases, the authors conclude that the SAC is compromised when all they do is look at budding or viability, assays that are not direct measures of the SAC. They make large conclusions that are not directly supported by the data. These conclusions need to be supported by experimental data or greatly toned down. Examples:

Fig 2c. They add nocodazole to bub mutants, but these mutants are not going to arrest so they don’t know what stage of cell cycle they are in yet make localization conclusions.

Fig 3A. One can't conclude SAC might be competent based on a budding assay. A Pds1 time course is required to make this conclusion. However, this point is semi-irrelevant since this experiment was already published in 2009 paper (Genes and Dev) from Biggins lab.

Fig 3D. They have no evidence that chromosome missegregation has anything to do with premature SAC silencing…Bub1 is important for biorientation so it is much more likely related to that than the SAC. It is confusing that the authors never mention the function of Bub1 in biorientation in this manuscript.

Fig 7. There is nothing in the figure about the SAC, it is all viability which could be due to some other function.

4) Gillet et al. (JCB 2004) previously addressed interdependency of checkpoint proteins for KT localization, this paper was not properly cited for these findings. Using ChIP, they found that Bub1 is dispensable for Bub3 binding at the centromere, which is in conflict with your microscopy findings. Please address this discrepancy experimentally (performing ChIP in your anaphase extracts to allay concerns) or textually.

5) The ndc80-7A mutant only shows a moderate decrease in Bub3/Bub1 anaphase localization compared to WT cells. The authors rightly conclude that this indicates Fin1 may be targeting other residues controlling Bub3/Bub1 localization. The authors should also textually include the complementary interpretation, which is that Ipl1 is targeting residues beyond the ndc80-7A mutant.

Minor points:

1) The manuscript should be explicitly clear as to where the ndc80-7A mutant comes from or what the residues are, as those unfamiliar with Ndc80 phosphorylation biology might not know what residues the authors are referring to.

2) The ndc80-7A mutant only shows a moderate decrease in Bub3/Bub1 anaphase localization compared to WT cells. The authors rightly conclude that this indicates Fin1 may be targeting other residues controlling Bub3/Bub1 localization. The authors should also textually include the complementary interpretation, which is that Ipl1 is targeting residues beyond the ndc80-7A mutant.

3) Is it possible that the increased persistence of Bub3/Bub1 in fin1 anaphase is due to the increased amount of Bub3/Bub1 at the KT from a metaphase localization defect as seen in authors’ previous work (Bokros 2016 Cell Report)?

4) Is there any in vivo evidence for Ipl1 phosphorylation of Ndc80 in anaphase?

5) Why is there a temperature sensitive effect on KT localization of Bub3/Bub1 in the fin1^ cells? Compare figure 4A/B to 1A

6) The title is vague/non-specific and too broad

Reviewer #3: This is an interesting manuscript that investigates the regulation of the spindle assembly checkpoint (SAC) during anaphase. The authors propose that the early release of Cdc14 in anaphase by the FEAR pathway results in the kinetochore recruitment of Fin1-PP1 in early anaphase which leads to Ndc80 dephosphorylation, thus disrupting the kinetochore association of the Bub1-Bub3 complex assuring SAC silencing. This is an important and often overlooked problem as most SAC research is focused on the metaphase to anaphase transition and therefore I consider this an important contribution. It is very difficult to find substrates of phosphatases and identifying Ndc80 as the relevant substrate for for PP1 in SAC regulation during anaphase is important.

All of the experiments are simple, straightforward and well presented. I have the following concerns about the data:

Figure 2.

Bub3 localization should be done in live cell imaging similar to Bub1 in Figure 1.

The interdependence of Bub1 and Bub3 for kinetochore localization is surprising. Does the failure to localize depend on Fin1? This should be tested.

Figure 3

SAC is active in Fin1-5A mutant with tagged Pds1 but tagged chromosomes missegregate.

The authors need an independent assay for chromosome missegregation in fin1-5A cells.

Figure 5

The data are hard to normalize as described because it is not clear what the "main band" is. The better normalization would be to Pgk1. Set the ratio of the hyperphosphorylated band to Pgk1 band at t=0 to 1.0

Figure 6

The ratio of Bub1-GFP and Bub3-GFP relative to Nuf2 M-cherry should be quantified in addition to the number of GFP positive cells.

minor concerns:

The strain list needs some corrections. I assume that supplemental table 1 has the relevant genotypes and not the complete genotypes. If that is the case, then indicate the background of all strains (W303a, S288C etc). The mating type locus should be uppercase (MAT) and the allele (a or alpha) should be bold.

The authors need to make clear that the experiments described at the bottom of page 8 are using plasmids in strains deleted for FIN1

There are several places where the writing is very "tentative" and the authors should be more conclusive.

For example:

"but untimely kinetochore localization of Fin1-5A may cause SAC silencing" is tentative and should be "but untimely kinetochore localization of Fin1-5A causes SAC silencing"".

"Fin1-PP1 may antagonize Ipl1 to regulate Bub1 localization at the kinetochore" is tentative and should be "We determined if Fin1-PP1 antagonizes Ipl1 to regulate Bub1 localization at the kinetochore".

"the data suggest that Fin1-PP1 likely antagonizes Ipl1 activity" is tentative and should be "the data suggest that Fin1-PP1 antagonizes Ipl1 activity".

Tentative phrases throughout the manuscript should be edited.

**Have all data underlying the figures and results presented in the manuscript been provided?**

Reviewer #1: Yes

Reviewer #2: Yes

Reviewer #3: Yes

PLOS authors have the option to publish the peer review history of their article (what does this mean?). If published, this will include your full peer review and any attached files.

Reviewer #1: **Yes: **Duncan J Clarke

Reviewer #2: No

Reviewer #3: **Yes: **Daniel Burke

---

## [Decision Letter · Decision Letter 1]

10 May 2021

Dear Dr Wang,

We are pleased to inform you that your manuscript entitled "Yeast Fin1-PP1 dephosphorylates an Ipl1 substrate, Ndc80, to remove Bub1-Bub3 checkpoint proteins from the kinetochore during anaphase" has been editorially accepted for publication in PLOS Genetics. Congratulations!

Yours sincerely,

Gregory P. Copenhaver

Editor-in-Chief

PLOS Genetics

Comments from the reviewers (if applicable):

Reviewer's Responses to Questions

**Comments to the Authors:**

Reviewer #1: The revisions to the MS address all of my previous comments. Indeed the authors have even done experiments that were suggested for future studies. The current MS provides important new insight and I fully support publication.

Reviewer #2: The authors have performed the experiments I requested and it has made the paper stronger.

Reviewer #3: This is a resubmission of manuscript that investigates the regulation of the spindle assembly checkpoint (SAC) during anaphase. The authors major conclusion is that the FEAR pathway regulates kinetochore recruitment of Fin1-PP1 in early anaphase which ultimately silences the SAC. They propose that Fin1-PP1 dephosphorylates Ndc80 which prevents the Bub1-Bub3 complex from associating with kinetochores. The data in the resubmitted manuscript support these major conclusions. It was my opinion before and remains my opinion now that this an important question and this paper represents a significant contribution to the field of SAC regulation. The authors have addressed all of the questions that I raised in my previous review, the manuscript has been well edited and is now ready for publication in PLOS Genetics.

**Have all data underlying the figures and results presented in the manuscript been provided?**

Reviewer #1: Yes

Reviewer #2: Yes

Reviewer #3: Yes

PLOS authors have the option to publish the peer review history of their article (what does this mean?). If published, this will include your full peer review and any attached files.

Reviewer #1: **Yes: **Duncan Clarke

Reviewer #2: No

Reviewer #3: No

**Data Deposition**

http://datadryad.org/submit?journalID=pgenetics&manu=PGENETICS-D-20-01796R1

**Press Queries**

---

## [Editor Report · Acceptance letter]

20 May 2021

PGENETICS-D-20-01796R1 

Yeast Fin1-PP1 dephosphorylates an Ipl1 substrate, Ndc80, to remove Bub1-Bub3 checkpoint proteins from the kinetochore during anaphase 

Dear Dr Wang, 

We are pleased to inform you that your manuscript entitled "Yeast Fin1-PP1 dephosphorylates an Ipl1 substrate, Ndc80, to remove Bub1-Bub3 checkpoint proteins from the kinetochore during anaphase" has been formally accepted for publication in PLOS Genetics! Your manuscript is now with our production department and you will be notified of the publication date in due course.

With kind regards,

Zsofi Zombor

PLOS Genetics

On behalf of:
